# Novel Targets for a Combination of Mechanical Unloading with Pharmacotherapy in Advanced Heart Failure

**DOI:** 10.3390/ijms23179886

**Published:** 2022-08-31

**Authors:** Agata Jedrzejewska, Alicja Braczko, Ada Kawecka, Marcin Hellmann, Piotr Siondalski, Ewa Slominska, Barbara Kutryb-Zajac, Magdi H. Yacoub, Ryszard T. Smolenski

**Affiliations:** 1Department of Biochemistry, Medical University of Gdansk, Debinki 1 Street, 80-211 Gdansk, Poland; 2Department of Cardiac Diagnostics, Medical University of Gdansk, Smoluchowskiego 17, 80-214 Gdansk, Poland; 3Department of Cardiac Surgery, Medical University of Gdansk, Debinki 7 Street, 80-211 Gdansk, Poland; 4Heart Science Centre, Imperial College of London at Harefield Hospital, Harefield UB9 6JH, UK

**Keywords:** heart failure, therapy, VADs, LVAD, mechanical unloading, myocardial recovery, cardiomyocytes, mitochondria, metabolism

## Abstract

LVAD therapy is an effective rescue in acute and especially chronic cardiac failure. In several scenarios, it provides a platform for regeneration and sustained myocardial recovery. While unloading seems to be a key element, pharmacotherapy may provide powerful tools to enhance effective cardiac regeneration. The synergy between LVAD support and medical agents may ensure satisfying outcomes on cardiomyocyte recovery followed by improved quality and quantity of patient life. This review summarizes the previous and contemporary strategies for combining LVAD with pharmacotherapy and proposes new therapeutic targets. Regulation of metabolic pathways, enhancing mitochondrial biogenesis and function, immunomodulating treatment, and stem-cell therapies represent therapeutic areas that require further experimental and clinical studies on their effectiveness in combination with mechanical unloading.

## 1. Introduction

Heart failure (HF) remains one of the most significant public concerns, affecting almost 100 million people worldwide, and 1-year mortality is estimated at 15–30% despite improved diagnostics and expanding knowledge [1]. The advanced stage of HF is characterized as progressive structural and functional heart deterioration, which causes increasing discomfort and severe dyspnoea in a patient, even without any physical activity [2]. Therefore, the main goal of pharmacological treatment is to reduce the main symptoms, thereby improving morbidity and mortality. For that purpose, angiotensin-converting enzyme (ACE) inhibitors, angiotensin II type I receptor blockers (ARBs), selected b-blockers (BB), aldosterone antagonists (AA), and inotropes might be recommended for patients with severe HF and reduced ejection fraction (HFrEF) [3]. Nonetheless, adjunctive HF medications provide only temporary relief; thus, non-pharmacological interventions play a more pivotal role in this field. Heart transplantation (HTx) is the last resort treatment, with a very high 1-year survival rate (90%) and long life expectancy after transplant (median 12.5 years) [4]. Although there are existing contraindications, low donor availability and immunosuppression-related side effects truly limit the therapeutic opportunity of this strategy [5]. In recent years, xenotransplantation became a promising future option for HF treatment before advances in genome editing with CRISPR-Cas tools. At the beginning of 2022, a group of researchers from the University of Maryland successfully performed a heart transplant from a line of genetically modified pigs to a human, extending the patient’s life for 2 months [6]. Given the early phase of development of this area and mentioned major limitations, ventricular assist devices (VADs) have been considered a reasonable alternative strategy in advanced HF, widely used as a bridge to heart transplantation or candidacy. Mechanical unloading supports the function of a failing heart and the perfusion of vital organs through reduced workload placed on the ventricles. In clinical practice, it might initiate a healing response with even myocardial recovery, making heart transplantation and mechanical support no longer required [7]. However, this phenomenon permitting device explantation is not fully understood, with its incidence varying in clinical practice, so the identification of underlying mechanisms attracts much interest and requires further studies. Therefore, this paper aims to summarize key pathways that lead to cardiac recovery and review current and prospective therapeutic approaches, including those based on mechanical unloading combined with pharmacotherapy and novel target treatments.

## 2. Mechanical Unloading of Failing Heart

### 2.1. Ventricular Assist Devices (VADs)

VADs are mechanical pumps that reduce the workload of the heart, providing support for the left ventricle (LVAD), the right ventricle (RVAD), or both ventricles (BIVAD). They are attached to the heart through cannulae that allow blood to enter a pumping chamber from which it is ejected into the systemic and/or pulmonary circulation. Devices are classified as pulsatile or continuous-flow based on how blood is pumped into the devices. The main goal of VADs is to restore cardiac output and stress relief before a failing heart leads to irreversible end-organ dysfunction, including pulmonary hypertension, renal failure, liver dysfunction, and cardiac cachexia. Currently, VADs might provide 1–2 weeks of support with ECMO and IMPELLA devices (short-term), 30 days by external centrifugal pumps (medium-term), or longer than 30 days via HeartMate3 and EXCOR pumps (long-term). Short-term VADs are intended for a wide range of clinical conditions, such as high-risk invasive coronary artery procedures for the management of cardiogenic shock, acute decompensated heart failure, or cardiopulmonary arrest. Among mid- and long-term VADs, LVAD is the most common type that has revolutionized the treatment of end-stage heart failure [8].

The first implantation of a mechanical assist device was performed in 1963 by bypassing the LV from the LA to the descending aorta at Baylor University in Huston. After 4 days, mechanical support was discontinued due to the patient’s coma and death [9]. Three years later, Dr DeBakey and Dr Liotta performed paracorporeal placement from the left atrium to the right subclavian artery [10,11]. This procedure was reported to be the first successful LVAD implantation as the patient recovered after 10 days of worsening HF and severe aortic insufficiency. It has been revealed that mechanical unloading immediately reduced LV pressure, improved cardiac output, and changed hemodynamic disturbances underlying the cardiac injury. Interestingly, LV assistance was acting salutary to systemic organ perfusion, accelerating myocardial recovery. Another milestone in LVAD’s history was HeartMate development, an electronically controlled assist device with Food and Drug Administration (FDA) approval for long-term use [12]. After the first generation of LVAD, pulsatile volume displacement pumps have been changed to continuous-flow rotary pumps (HeartMate-II). In the current HeartMate-III device, centrifugal pumps with an impeller provide better cardiac support, a wider range of flows, and a lesser risk of device malfunction. Moreover, the size and noise were reduced, materials are lighter and more durable, and proposed modifications, such as better surgical technique [13], mobile phone application for self-care [14], and exercise training [15], improved patients’ outcomes and life quality. Interestingly, many clinical trials are still finding novel solutions for adverse events and health status monitoring. This meaningful evolution makes LVAD implantation easier and safer, significantly improving long-term survival, with patients living a decade with mechanical support [16].

### 2.2. The Current Status of Left Ventricular Assistance Devices (LVADs) Therapy in Heart Failure

Over the past decade in the United States, 25,551 patients with heart failure underwent continuous-flow LVADs placement [17]. According to the 2021 European Society of Cardiology (ESC) Guidelines, LVAD implantation may be recommended when symptoms persist, despite optimal medical treatment, and the absence of severe right ventricular dysfunction and/or tricuspid regurgitation [3]. Furthermore, a potential patient must have at least one of the following: LVEF < 25% and unable to exercise or able but with peak VO2 < 12 mL/kg/min and/or <50% predicted value, ≥3 HF hospitalizations without an obvious precipitating cause (during 12 months), dependence on inotropic therapy or temporary MCS, and progressive end-organ dysfunction. Great emphasis is also put on stable psychosocial background and support from society. Living alone and having poor overall mental health is a contraindication to the same extent as infection, severe renal dysfunction, and ventricular arrhythmias. Currently, there are three therapeutic goals for LVAD implantation: bridge to transplantation (BTT), bridge to candidacy (BTC), and destination therapy (DT). In the BTT regimen, the device supports the patient and improves physiology until a donor becomes available. In turn, BTC provides time for the patient to become eligible for cardiac transplantation by, for example, decreasing body mass index, achieving a required cancer-free period, or securing financial and family support. The last alternative option is DT for patients with end-stage HF and transplantation contraindications, which allows for discharge from the hospital and improves the quality of life.

The annual report from Interagency Registry for Mechanically Assisted Circulatory Support (Intermacs) has shown that DT is currently the most common form of therapy [17]. The importance of BTT has decreased and become the rarest therapeutic target (8,9% of total implantations). The advantage of long-term use started in 2015 (nearly 50% of total implantations), reaching its peak in 2019 (73,1%) [17,18]. Interestingly, the Randomized Evaluation of Mechanical Assistance for the Treatment of Congestive Heart Failure (RE-MATCH) showed that patients randomized to receive LVAD as DT lived longer in better health in comparison to the subjects with optimal pharmacological treatment [19]. Moreover, the main cause of death was not terminal HF as in the medical group but systemic infection and device malfunction.

### 2.3. Bridge to Recovery: A Meaningful Phenomenon in LVAD Therapy

As previously mentioned, short-term mechanical unloading is recommended to prevent cardiogenic shock (CS) in acute conditions [3]. It has been shown that temporary LVAD implantation haemodynamically stabilized patients during acute cardiac decompensation. This strategy has been called the bridge to recovery or decision (BTR/D) and in global recommendations is referred to be keeping a patient alive until cardiac function recovers or enables time for clinical decisions on long-term management to be made [20]. In BTT, the duration of mechanical support remains naturally longer due to the limit of heart donors and the change in the urgent cardiac recipient status to the chosen recipient. This entails an increasing number of LVAD patients waiting longer and longer for transplantation. However, it has been reported that patients with prolonged mechanical support improve major hemodynamic parameters, quality of life, and health status, which are preserved to the moment of heart transplantation [21,22,23]. Moreover, LVAD affects salutary to organ perfusion; thus, post-transplant patients achieve better outcomes and life expectancy. Therefore, “recovering” might also refer to advanced conditions since the reversal of cardiac dysfunction has been frequently reported. Frazier et al. suggested that mechanical-assisted recovery might provide long-term survival without transplantation and even a supporting device [24,25]. It has been shown that 4/5 selected patients with severe congestive heart failure survived LVAD removal (one died of noncardiac-related causes) and, at the time of publishing, were alive and well 35, 33, 14, and 2 months [25]. This established myocardial recovery as sustained normalization of LV function in patients previously assigned to BTT but followed by LVAD explantation. Nowadays, there is a lack of standardization and clear criteria for patients weaning from the device. Monteagudo Vela et al. proposed that LVAD removal might be considered if a patient is in New York Heart Association (NYHA) class I and has a normalized ejection fraction > 40%, cardiac index > 2.4 L/min and peak oxygen intake > 50% predicted. However, guidelines standardization for less invasive LVAD explantation and weaning protocol might reduce surgical complications and preserve myocardial recovery, leading to longer survival [26]. Given the various weaning protocols, different percentages of patients who met the criteria are also observed. Over the past two decades, the average of patients who successfully underwent LVAD explantation was 12.5% [27,28,29,30,31,32,33,34]. Selzman et al. highlighted that the younger population (<40 years old), with non-ischemic origins of HF, has a higher likelihood of myocardial recovery and faster device removal [35]. However, it should be emphasized that LVAD therapy is associated with long-term survival free from recurrent heart failure, which is rarely achieved with any current HF pharmacotherapy [34].

A growing body of evidence has shown that, in respondent patients, LVAD induces structural and functional changes at the cellular, molecular, and whole-heart levels, known as reverse remodelling [36,37,38]. The cellular processes are thought to be more profound and more significant than the changes observed in cardiac function. The potential mechanism of cardiomyocyte remodelling is irreversibly connected to its stretching reduction (Figure 1). Hemodynamic support stimulates karyokinesis and favours the ability to divide, which was confirmed by increasing diploid cardiomyocytes in myocardial samples [39]. An increased number of circulating progenitor cells might correlate to ongoing cardiac recovery; however, their number seems to be transient over time [40,41]. Particular attention has been paid to the specific gene expression in human unloaded hearts, such as expression of profibrotic, contractile, involved in Ca^2+^ cycling, and proinflammatory proteins [42,43,44]. Reduced level of cytokines was found both in serum and myocardial tissue in patients with improved cardiac function after LVAD implantation compared to the non-responder group [45]. Interestingly, the signal transducer and activator of transcription 3 (STAT3) was responsible for modulating the immune response. Moreover, pre- and post-intervention levels of cytokine were correlated with further LV improvement, suggesting inflammation is an essential factor of LVAD response. It has been highlighted that levels of cytokines in the myocardium, especially tumour necrosis factor (TNF), might predict patients’ recovery [45,46]. The systolic function improvement is thought to be initiated by preserving the abundance of key regulatory proteins (sarcoplasmic reticulum calcium adenosine triphosphatase, SERCA) and a decrease in the Na^+^/Ca^2+^ exchanger (NCX) during LVAD therapy [47]. It results in greater calcium uptake and contributes to greater cardiac contractility [48]. Furthermore, mechanical unloading has improved Ca^2+^ handling through significant tubule remodelling [49]. It has been shown that the density and activity of L-type Ca^2+^ channels and transverse tubules (t-tubule) have been normalized in the rodent model of mechanical unloading compared to unfavourable outcomes from only HF rats. T-system defects and related-Ca^2+^ handling aberration are features of heart failure progression and, hence, are thought to be the key to the proper functioning of cardiomyocytes and novel predictors for functional cardiac recovery after mechanical unloading [49,50]. Mechanical support also plays an ambiguous role in extracellular matrix (ECM) remodelling. Lower profibrotic gene expression might contribute to reduced collagen content [51,52]; however, some studies indicated increased fibrosis in heart samples after LVAD support. It can be explained by the decreased breakdown (through decreased activity matrix metalloproteinases activity, MMPs) and increased synthesis of collagen (via increased activity of angiotensin I and II; Ang), which were reported [53,54]. However, ECM turnover is highly related to the aetiology of the injury, RV function, patient’s age, or type of LVAD support, making the influence of mechanical support difficult to determine [55]. Metabolic changes and cellular pathways play a pivotal role in reverse remodelling. It has been reported that effective hemodynamic support induces glycolysis and increases glycolytic metabolites without directing them through the tricarboxylic acid cycle (TCA) [56]. To provide an alternative energy source, the increased level of amino acids was found as a compensatory mechanism. Based on the heightened level of cytosolic pyruvate and lactate, Diakos et al. suggested that glycolysis–pyruvate oxidation mismatch is a result of impaired mitochondrial function or protective mechanism, leading to oxidative stress reduction. Mitochondrial volume density and mitochondrial DNA (mtDNA), although significantly lower at implantation time compared to the healthy control, have slightly increased during mechanical unloading. Similar results were confirmed in further study, where up-regulated glycolysis initiated activation of protective pathways, such as the pentose phosphate pathway and 1-carbon metabolism in post-LVAD responders [57]. This specific mechanism protects cells against reactive oxygen species (ROS) and increases the synthesis of nucleotides. Furthermore, restoration of the pyruvate–lactate axis was recently highlighted as a predictor of myocardial recovery [58]. It has been reported that increased myocardial expression of mitochondrial pyruvate carriers (MPC) was observed in human myocardial samples of patients who respond to mechanical unloading and improved cardiac function. The non-responding group had lower MPC1 levels before and after LVAD implantation and finally underwent heart transplantation due to a failing heart condition. Moreover, loss of MPC activity, which was resulting from the pyruvate-lactate axis imbalance, promoted hypertrophy and HF development by targeting lactate to the extracellular space, which was confirmed both in in vitro (H9c2 cells) and in vivo models (MPC1-deficient mouse). In turn, Diakos et al. revealed that mechanical unloading does not induce hypertrophy regression towards atrophy based on cardiomyocyte size, glycogen content, myocardial mass, and proatrophic gene expression [59]. However, some studies observed up-regulation of the proteolysis pathway in heart samples at the time of LVAD explantation [60,61]. Overall, mechanical support causes cascades of reactions in which gene expression, proliferation, apoptosis, fibrosis, immune response, and cardiomyocyte metabolism are modulated. Some of these changes provide prognostic value, and others are the reason why not all LVAD patients achieve myocardial recovery. Currently, the phenomenon of reverse remodelling might be examined in both in vitro [62] and in vivo models [63]. Therefore, further studies correlating cardiac function with cellular disturbances might be useful in designing novel target therapies and increasing the number of patients successfully weaned.

### 2.4. LVAD Limitations

Limitations regarding myocardial recovery might be divided into two categories: adverse events and unfavourable changes in cardiomyocytes, both associated with prolonged mechanical unloading. In the first case, LVAD patients struggle with many post-implantation complications, as shown in Table 1, with bleeding being the most frequent but not the most dangerous [64]. Multisystem organ failure (16.4%), stroke (15.6%), heart failure (12.5%), and major infection (5.7%) cause half of the deaths in the LVAD population [17]. Continuous flow is thought to increase the risk of gastrointestinal bleeding, lack of pulse, and thromboembolic complications [65]. Despite technological progress and novel medical solutions, adverse events, such as bleeding, right heart failure, and infections, continue to be limiting factors in sustained recovery. LVAD patients should routinely assess predictor factors by echocardiographic assessment and biochemical parameters in their referring centre, which has become difficult due to the COVID-19 pandemic [66]. Noteworthily, LVAD is not fully implantable, and the external system controller with a driveline protruding from the patient’s abdomen might negatively affect physical and mental health, requiring the patient’s acceptance. Secondarily, the prolonged hemodynamical support might produce an “atrophic”, proinflammatory, and profibrotic response, with mitochondrial dysfunction as well (Figure 1). Therefore, clinical improvement might occur early during LVAD or not, with/without time regression [29]. Overall, mechanical unloading is speculated not to be sufficient to achieve total heart recovery without amelioration of detrimental factors. Thus, the development of a novel target treatment is urgently needed.

## 3. Combination of Mechanical Unloading and Pharmacotherapy for Chronic Heart Failure Treatment

### 3.1. Potential Benefits

The effectiveness of LVAD therapy is evidenced by a sufficient myocardial recovery that allows device explantation and life without mechanical support [75]. To increase the number of patients who achieved those goals, combination therapy was first proposed in 2001 to “maximize the efficacy of LVAD as BTR” [76]. The basis of combination therapy has become the drive to reverse cardiac remodelling, followed by the stimulation of physiological cardiomyocyte growth. This formed the “Harefield protocol” in which LVAD therapy was enlarged by clenbuterol [77]. After 5 years, the Harefield protocol was tested as a combination of mechanical unloading with specific drug therapy for 15 patients with severe HF due to nonischaemic cardiomyopathy [78]. In the pharmacological regimen, four HF medications reduced LV remodelling, and then clenbuterol was administered to prevent myocardial atrophy, which had been proven in prior studies [79,80,81]. The examination showed significant cellular and functional improvement, which translated into a high rate of survival and recovery with combination treatment (Table 2). This is of particular note given that, at that time, LVAD therapy was associated with only a small percentage of recovery sufficient for device explantation (5% [27], 8% [28]) and an equal chance of 1-year survival [19]. Further, the same strategy was enlarged to 20 patients with dilated cardiomyopathy and demonstrated great utility and effectiveness [35]. Concentrating on survival and durability of recovery, Birks et al. demonstrated a significantly high rate of heart failure reversal (Table 2). Particularly interesting is the fact that 10/12 patients weaned from the device survived, remaining in NYHA class I with cardiac normalization and sustained recovery.

The promotion of reversed remodelling has also been described in another pharmacologic regimen. In the study of Grupper et al., patients who had ischemic cardiomyopathy were randomly assigned to the neurohormonal blockade (NHB) therapy (receiving ACE, ARB, BB, or AA) or the control group without any NHB drug after LVAD implantation [82]. Achieving progressive normalization, the NHB patients experienced greater myocardial recovery and down-regulation of neurohormones after 6 months (Table 2). Furthermore, Grupper et al. have also demonstrated improvement in NYHA classification (*p* = 0.024) and reduced hazard of death and HF-related hospitalization (*p* = 0.013) in response to combination therapy. These changes are in line with previous [83] and recent [84] studies. Patel et al. revealed the reversibility of advanced HF by combined LVAD support with maximal adjunctive HF medications (lisinopril/losartan, carvedilol/metoprolol, and spironolactone). Positive results of the three-step testing (echocardiography, exercise, and right heart catheterization) allowed three patients with idiopathic dilated cardiomyopathy (IDCM) to withdraw mechanical support. Overall, 20% of coronary artery disease or IDCM patients achieved biventricular recovery alongside decreased LV mass (*p* < 0.001), LV internal diastolic diameter (*p* = 0.003), and increased LV ejection fraction (*p* < 0.001) compared to pre-implantation status (Table 2). Interestingly, histopathologic changes exposed a significant reduction in myocardial fibrosis and hypertrophy, proving molecular and cellular response to mechanical support. In turn, the confirmation of improved health status and life expectancy was presented in the observation study by McCullough et al. (Table 2). Analysing outcomes from INTERMACS (2008–2016), it has been revealed that LVAD patients receiving NHB therapy (singular or plural combination of ACE/ARB, BB, and mineralocorticoid receptor antagonist, MRA) have longer survival estimates (*p* < 0.001) and better quality of life defined by the Kansas City Cardiomyopathy Questionnaire (KCCQ; *p* = 0.02) and 6-minute walk test (ft; *p* < 0.001). Moreover, recovery permitting device explantation was also more favourable in the NHB group (*p* = 0.04). LVAD patients receiving triple NHB therapy performed best, analogical to data from HFrEF guidelines (without mechanical support). The consistency in results from small clinical studies and cohort analysis gives credence to the notion that intensive NHB treatment is beneficial for LVAD patients and may set the trajectory for future medical regimens.

A different strategy might be the pre-operative use of sacubitril–valsartan (ARNI: angiotensin receptor antagonist and neprilysin inhibitor) to reduce post-operative mortality in patients undergoing LVAD implantation. Heder et al. reported an association with better survival outcomes after cardiac surgery (LVAD or HTx) [85]. According to Kaplan and Meier’s analysis, the lowest rate of death was observed in ARNI (13.6%), then ACEi, ARB (19.4%), and, lastly, the no-vasoactive group, with the worst 30-day survival rate (62.5%; *p* = 0.043). It has been previously reported that angiotensin–neprilysin inhibition reduces NT-proBNP levels in patients with acute HF, with no greater incidence of renal dysfunction, hyperkalaemia, and symptomatic hypotension [86]. Interestingly sacubitril–valsartan is currently recommended in HFrEF to reduce the risk of HF hospitalization and death [3], and its therapeutic opportunity is currently enlarged to LVAD support (Table 3). In two randomized clinical trials, particular emphasis will be placed on medication-related adverse events, mortality, and cardiac improvement, as well as to confirm the safety and utility of this adjuvant strategy.

A combined approach was also examined in a rodent model of HF using the established and reproducible procedure of LV unloading [87]. Since chronic support might cause alteration in Ca^2+^ cycling, increase fibrosis, and induce myocardial atrophy [88,89], Navaratnarajah et al. selected ivabradine (Iva) as a proven agent to counteract these deleterious effects [87]. It has been reported that 4 weeks of combined treatment was successful in two of three key determinants but did not prevent myocardial atrophy. Proposed co-therapy improved cardiomyocyte contractility, increased Ca^2+^ transients, and normalized collagen area fraction compared to only unloaded hearts. Thus, the combination treatment promoted to a greater extent reverse remodelling. Metoprolol has also been investigated but only reduced myocardial atrophy. A similar study was performed to examine the effects of clenbuterol and metoprolol in the same HF model during mechanical unloading [90]. Unfortunately, these two agents were not effective in preventing cardiomyocyte atrophy and enhancing calcium handling.

In summary, The Harefield protocol was one of the first studies to successfully combine pharmacological agents with LVAD. Particularly interesting are higher rates of recovery when adjuvant medical therapy is included in the LVAD strategy (Table 2). Therefore, this strategy could modulate the detrimental effects of prolonged mechanical unloading and promote reverse remodelling. In addition, the clinical success raised patients’ hope for cardiac recovery and device explantation. However, the main limitation in many mentioned studies is the relatively small number of patients and the lack of a control group, making the influence of pharmacological agents in the remodelling process difficult to determine. Hence, the exact mechanism initiating patients’ recovery is scarcely examined and requires large-scale and well-designed clinical trials. The aggressive medications lower myocardial energy demand by reducing hemodynamic instability and modulating neurohormonal response; however, the other side of cardiac metabolism associated with cell proliferation, substrate oxidation, and mitochondrial function attracts much interest [91,92].

### 3.2. Novel Pharmacotherapies for Cardiomyocyte Regeneration during LVAD Support 

#### 3.2.1. Stimulation of Cardiomyocyte Proliferation 

Mechanical unloading is associated with cardiomyocyte atrophy, resulting in a decline in myocyte densities and LV mass continuously [60,93]. The hypertrophy regression is speculated to attenuate heart recovery, but it is not fully proven whether it is related to mechanical unloading or the pathophysiology of HF itself [94,95]. However, a few clinical strategies in which cardiomyocyte proliferation is successfully stimulated might be beneficial for durable cardiac recovery. 

Regenerative therapy is based on embryonal or adult stem cells as an effective therapeutic target for cardiogenesis and angiogenesis. The main goal of this strategy is to boost cell production directly and then indirectly to improve cardiac function by injecting stem cells into a wounded region of the heart [96]. In prior experimental studies, the function and volume of LV, as well as the size of the infarction, have been improved after receiving millions of potential heart cells [97,98,99]. Mesenchymal precursor cells (MPCs) have shown cardioprotective, pro-vascular, and anti-inflammatory properties [100,101,102]; thus, they are considered to be a promising therapeutic agent for chronic HF [100,103]. In turn, clinical trials are not entirely consistent on whether stem-cell-based therapy could promote cardiac recovery in HF [104,105] or the LVAD population only [106,107,108]. Stempien-Otero et al. reported that injection of bone-marrow-derived cells (bone marrow mononuclear cells; BMMNCs, CD34+, and CD34–) has not counteracted myocardial fibrosis and inflammation after LVAD implantation [106]. Left ventricular recovery due to MPCs injection was not reported by Yau et al. either [107]. In contrast, the potential signal of efficacy was claimed in the study of Ascheim et al. in which the likelihood of successful temporary LVAD weaning and survival after 3 months was higher in the stem cell (MPC) group than in the control group [108]. Overall, stem-cell-based therapies are an encouraging approach; however, the different microenvironments of injury, selection of optimal cell types, methods of delivery, therapeutic doses, and cell quality (age-dependent) are the most challenging, hence requiring more clinical validations [96,109]. Therefore, a strong need arose for clinical trials in the mechanical unloading area, as shown in Table 3.

Since microRNAs (miRNAs) were reported as key regulators of cardiomyocyte proliferation, a few studies investigated the possibility to promote cardiomyocyte division and lead heart regeneration by injecting specific RNAs. It has been discovered that the administration of hsa-miR-590 and hsa-miR-199a and, further, miR-19a/19b have enhanced cardiomyocyte proliferation and normalized cardiac function parameters in animal models [110,111]. Moreover, the circulating miRNAs are considered to be promising new biomarkers, prognostic tools, and indicators of response to HF treatment [112]. Despite the lack of clinical studies, potential therapeutic opportunities in miRNA expression are currently examined after long-term mechanical support and suggest significant deregulation of vascular remodelling, possibly related to endothelial dysfunction [113]. In turn, Sansone et al. have indicated that LVAD support acts favourably to microvascular perfusion, but the lack of pulsating flow can lead to endothelial activation and its residual dysfunction. Moreover, reduced NO-dependent vasodilation might also contribute to a worse cardiovascular outcome [114]. Therefore, the continuous speed changes are thought to have a beneficial effect on the endothelium, thereby on greater cardiac and organ outcomes, but the relative youth of this area is highlighted (NCT04539093, ClinicalTrials.gov, accessed on 10 June 2022). Expanding knowledge in this field is aimed to find potential therapeutic targets and limit negative changes caused by prolonged circulatory support. 

#### 3.2.2. Regulators of Cardiac Substrate Metabolism

Cardiac metabolism plays a central role in the pathophysiology of HF [115]. Its alterations profoundly impair cardiac function and develop further progression. Metabolic modulators are recently thought to display therapeutic potential in cardiovascular diseases [92,116]. The reduction in fatty acid (FA) oxidation is one of the starting points. In normoxia, FA breakdown is the main source of ATP (about 70%), alongside glucose and lactate [117]. It should be noted that energy production from glucose requires less oxygen than FA, which is crucial in ischemia conditions [118]. Therefore, decreased FA metabolism has been reported in many animal models, also related to left ventricle dysfunction, and it is mainly observed not in an early but advanced stage of HF [119,120,121]. It might be caused by the lower energy demand and optimized ATP production or, as previously mentioned, directing glucose into alternative and more beneficial pathways (Figure 1). Thus, the new therapeutic approach for LVAD patients might be the regulation of FA oxidation in favour of glycolysis to generate biomolecules promoting the normal energy state of cardiomyocytes [109].

Perhexiline, a drug developed for angina, was proven to be simultaneously favourable for hemodynamic and metabolic cardiac function in cardiovascular disease [122,123,124]. It raises glucose utilization and decreases beta-oxidation through inhibition of fatty acids mitochondrial transporter (carnitine palmitoyltransferase-1) [122,125]. It has been shown that perhexiline increases the energetic state (PCr/ATP ratio; 1.16 to 1.51) and left ventricular systolic function (LVEF; 24% to 34%) in HF [122,126]. Moreover, perhexiline-treated patients have significantly improved NYHA classification (*p* = 0.036) and quality of life after 1 month of treatment compared to the non-treated group [126]. Recently, perhexiline has been investigated for hypertrophic cardiomyopathy (HCM) treatment as a key regulator of cardiomyocytes’ energy balance (NCT04426578, ClinicalTrials.gov, accessed on 10 June 2022) 

Another mitotrope, trimetazidine, is also thought to have potential cytoprotective effects in HF treatment [127]. It might enhance cardiac function by preventing cardiomyocytes from many deleterious events, such as apoptosis, fibrosis, and inflammation [128,129,130]. Moreover, Tuunanen et al. highlighted that trimetazidine, to a greater extent, improved whole-body insulin sensitivity and glucose control than decreased FA oxidation as the main mechanism of potential improvement. However, even a slight lowering of FA breakdown plays a significant role in the Randle cycle and increases the glycolytic feed into pyruvate dehydrogenase [131], which is still a significant benefit to LVAD therapy. This is in line with the study by Cluntun et al., suggesting that an increase in MPC subunits, as well as inhibition of the cellular lactate exporter mono-carboxylate transporter 4 (MCT4), could represent an interesting therapeutic approach [58]. Therefore, trimetazidine has been clinically investigated, and an increase of EF by 3.9% [132] or 7% alongside 17 mL left ventricular end-systolic volume decreasing has been reported [133]. However, it should be emphasized that those studies included only small cohorts of HF patients (*n* = 19 and *n* = 55, respectively) and well-designed clinical trials on a bigger scale have not been performed yet. Given insufficient clinical evidence, trimetazidine is not included in the latest ESC Guidelines (2021) for chronic HF treatment [3].

Sodium/glucose cotransporter 2 (SGLT2) inhibitors are thought to be promising cardioprotective agents in cardiovascular diseases. Empagliflozin, the very first tested, was revealed to reduce the risk of HF hospitalization in patients with and without diabetes type 2 [134,135]. Further clinical trials with canagliflozin and dapagliflozin established a reduction in cardiovascular death or HF hospitalization [136,137]. However, the cardioprotective mechanism, which distinguishes SGLT2 inhibitors from other glucose-lowering drugs, is not fully elucidated. One controversial thesis is an optimization of cardiac energy metabolism by reducing blood glucose levels with simultaneously increasing ketones synthesis and providing “super fuel” for the heart [138,139]. In opposition, SGLT2 inhibitors are speculated to not only regulate glucose concentration but also homeostasis and associated parameters (blood pressure, haematocrit, and sodium level) [140,141,142]. All these changes act favourably on circulatory unload and ventricular stress, reducing the likelihood of cardiovascular death. There are many noteworthy clinical trials examining SGLT2 inhibitors, which shows their great therapeutic potential not only in the cardiovascular field. Since volume overload impairs right ventricle function, leading to hypertrophy and fibrosis [143,144], these specific inhibitors might prevent right ventricular failure and promote myocardial recovery after LVAD implantation. Counteraction of hemodynamic disturbance will be tested in the novel clinical trial shown in Table 3.

The selective agonists of peroxisome-proliferator-activated receptors (PPARs) regulate the expression of specific genes in lipid or glucose metabolism [145,146]. The PPARs family belongs to three isoforms of nuclear receptors (PPARα, PPARβ/δ, and PPARγ) with different locations, biological effects, and medical purposes. Fibrates (PPARα agonists) are most known for atherogenic dyslipidaemia treatment due to lowering triglycerides (TG) and, to a lesser extent, changing high- and low-density lipoproteins levels [147]. They are highly expressed in skeletal muscle, heart, liver, and brown adipose tissue, which increases FA uptake, oxidation, and transformation to TAG [148]. PPARβ/δ is even more extensively expressed, thereby controlling many metabolic pathways or even mitochondrial biogenesis [149,150,151]. In turn, PPARγ is strongly activated in adipose tissue, macrophages, cardiomyocytes, vascular smooth muscle, and endothelial cells [149,152]. Moreover, the agonists are involved in adipocyte differentiation, which lowers the number of cardiac cytotoxic lipids and glucose metabolism by increasing insulin sensitivity [153]. Therefore, PPARγ agonists are recommended for diabetic patients at high risk of HF [154]. Overall, PPAR agonists are considered to prevent cardiomyocytes from ATP depletion and enhance mitochondrial biogenesis, important therapeutic targets in LVAD remodelling (Figure 1) [148,153]. Treatment with rosiglitazone (PPARγ agonists) has simultaneously improved LV diastolic function and reduced myocardial fibrosis in a rat model of diabetes [155]. Improvement in cardiac function was also observed in human and animal models of diabetes after pioglitazone (another PPARγ agonist) administration [156,157]. However, there is still a lack of clinical trials analysing cardiac outcomes in other states than diabetic cardiomyopathy. Montaigne et al. indicated that a selective PPARβ/δ agonist or a dual PPARα–PPARβ/δ agonist might boost mitochondrial function, particularly in the early stages of HF remodelling [153]. Given the differential functions in cardiac metabolism, more research is required to investigate the beneficial effect of using singular or plural PPAR agonists, which has recently been highlighted in many studies [153,158].

In summary, the regulation of cardiac substrate metabolism is an important approach with still few translational trials compared to any other pharmacological agents. It targets cellular and molecular pathways with a proven global effect on the heart, such as FA oxidation, glucose metabolism, ATP production, and even gene transcription. It is underlined that heart recovery is inseparable from specific metabolic changes that not only increase energy production but also activate cardioprotective pathways. Hence, this new therapeutic strategy should be enlarged for LVAD patients, where specific metabolic changes have been recently reported (Figure 1). In this regard, particular attention should be paid to perhexiline and trimetazidine, which improves the energetic state of the heart, SGLT2 inhibitors by homeostasis regulation and reduction in ventricular loading, and, finally, PPAR agonists with benefits to mitochondrial biogenesis.

#### 3.2.3. Mitochondria-Targeted Treatment

Mitochondrial biogenesis, structure, and function have been of special interest with wide clinical testing since their improvement is thought to ameliorate cardiac function directly [159]. The reduction in pathological ROS production is one of the therapeutic targets and is believed to restore energetic balance in cardiomyocytes. To find the optimal antioxidant, supplementation of coenzyme Q (CoQ) was proposed for chronic HF patients in NYHA classification III or IV [160]. CoQ plays a significant role in the mitochondrial electron transport chain as an electron acceptor, thereby contributing to ROS reductions. At week 106 of supplementation, the outcomes were positive and treated patients showed a significantly lower risk of cardiovascular deaths (*p* = 0.026) and HF hospitalization (*p* = 0.033) in comparison to the placebo group. These results are in line with a more recent meta-analysis, where reduced mortality and improved exercise capacity were noted in HF patients with CoQ supplementation [161]. Moreover, the CoQ with better mitochondrial bioavailability (MitoQ) was reported to restore mitochondrial respiration and membrane potential in an animal model of heart failure induced by pressure overload [162]. Another rationale for therapeutic use might be decreased endogenous synthesis of CoQ with age [163] and its efficacy in lowering proBNP and improving cardiac systolic function in an elderly population [164]. In turn, the Szeto–Schiller (SS) peptides, especially SS-31, have demonstrated protective properties to cardiolipin, maintaining electron carrying function and ROS utilization [165]. SS-31 safety and toleration have been accepted in two clinical trials, whilst a single infusion in a high dose was beneficial for LV volume [166] but, in the long-term, did not decrease myocardial infarct size [167]. Decreased content of cardiolipin and its mitochondrial decomposition was reported in the myocardium after LVAD support, suggesting that cardiolipin is a potential therapeutic target [168]. In addition, ROS scavenging might be another promising approach when its elevated levels and correlating mitochondrial dysfunction were found in patients without cardiac response during LVAD therapy [57]. However, the effectiveness of the antioxidative approach in enhancing mitochondrial function remains to be established in clinical practice [169].

To prevent pathological heart remodelling, it is proposed to maintain the nicotinamide adenine dinucleotide (NAD+) pool and NADH (reduced form of NAD+)/NAD+ ratio [170]. An elevated ratio of NADH/NAD+ with cytosolic protein hyperacetylation, including malate–aspartate shuttle proteins and oligomycin-sensitive conferring protein in ATP synthase complex, contributed to the worsening HF in humans and in an animal model of mitochondrial complex-I deficiency. Furthermore, Lee et al. highlighted that elevating the NAD+ level might normalize redox status and improve cardiac function, predicting the high translational potential of the NAD+ precursors. A heightened level of intracellular NAD+ was observed in murine-immortalized heart endothelial cells (H5V line) after 24 h incubation with fatty acids (linoleic acid and docosahexaenoic acid) and atorvastatin [171]. In blood, nicotinamide riboside (NR) was confirmed to successfully increase the level of circulating NAD+ in healthy human voluntaries without serious side effects [172], unlike niacin supplementation [173]. Therapeutic opportunities of NR were recently investigated in HF patients undergoing LVAD implantation (Table 3) and showed that oral administration was associated with reduced pro-inflammatory activation [174]. However, results relating to mitochondria status or myocardial recovery have not been published. Ongoing clinical trials (NCT04528004, NCT03423342, ClinicalTrials.gov, accessed on 10 June 2022) will evaluate whether boosting NAD+ levels through NR administration may improve cardiac function in heart failure. Despite the lack of studies evaluating the NAD+ pool after mechanical support, it should be emphasized that compounds known to ameliorate mitochondrial structure and function might induce reverse remodelling and thus warrant further investigation.

#### 3.2.4. Inhibition of Inflammation

The anti-inflammatory approach originated from the strong need to counteract high mortality and morbidity, along with a better understanding of protective or harmful immune mechanisms in heart failure progression [175]. It has been reported that levels of circulating cytokines, such as interleukin (IL) 1, 2, or 6, and TNF, are increasing simultaneously with the worsening status of HF patients [176,177]. It might be caused by LV dysfunction (via hypertrophy and further fibrosis) and endothelial dysfunction (through apoptosis and reduced NO synthesis), overall contributing to myocardial failure [178]. Hence, lowering the level of proinflammatory cytokines is thought to initiate a healing response.

Patients with LVAD might be at particular risk of a higher level of serum TNF-α levels compared to only HF or HTx subjects [179]. Tabit et al. suggested that elevated levels of TNF-α directly stimulate thrombin-induced angiopoietin-2 (Ang2) expression and jointly induce pathological angiogenesis, leading to angiodysplasia and increasing the risk of non-surgical bleeding. Therefore, TNF-α blockade could prevent LVAD patients from high risk of these complications. In a prior study, TNF-α inhibition failed to show improvement in symptoms and quality of life whilst increasing the risk of hospitalization for worsening heart failure [180]. In contrast, administration of pentoxifylline has improved cardiac function (LVEF increased by 32%) and reduced markers of inflammation (CRP, TNF-α) with a preserved pool of circulating TNF in ischemic cardiomyopathy [181]. It is noteworthy that TNF might improve post-ischemic functional recovery; therefore, the use of strong inhibition reduces the cardioprotective effect [182].

Anakinra is a recombinant human receptor for IL-1 and blockades successfully both isoforms (α/β) from proinflammatory signalling. Concordant with prior studies, IL-1 inhibition has effectively dealt with an inflammatory response in acute myocardial infarction [183] and LV dysfunction [184]. Moreover, lowering IL-1 levels is believed to boost cardiac function by restoring calcium handling and preventing cardiomyocytes from abnormal contractility and hypotrophy [185]. Therefore, Anakinra is also considered to be an adjuvant agent in the LVAD area (Table 3). The first outcomes from the clinical trial have convincingly demonstrated therapeutic potential [186]. Healy et al. have shown not only CRP reduction (about 76%) but also a 67% increase in EF after 6 months of a short course of Anakinra (2 weeks). Notably, the pilot study did not include a control group, so it is not possible to separate the influences of mechanical unloading from immunomodulating treatment. However, the efficacy and safety of LVAD combination with IL-1 blockade have been initially confirmed and encourage future well-designed clinical trials.

#### 3.2.5. Other Strategies for Cardiac Regeneration

Restoration of SERCA2a expression is thought to prevent severe systolic and diastolic dysfunction equally to mechanical assist devices in advanced HF treatment [187]. Down-regulation of SERCA2a was observed in failing heart samples, which translated to dysregulation in Ca^2+^ homeostasis and impaired myocardial contractility through increased intracellular calcium concentrations [188]. Therefore, therapy based on SERCA2a gene delivery was proposed to reverse HF progression as a novel modality for treatment. A signal of potential opportunities to reduce the number and recurrence of cardiovascular events has been described following a single dose of adeno-associated virus serotype 1 (AAV1) vector carrying SERCA2a (AAV1/SERCA2a) in patients with advanced heart failure [189]. However, the combination of gene therapy with mechanical circulatory support did not show positive outcomes in the SERCA-LVAD trial due to the small cohort of patients and safety concerns [190]. In turn, istaroxime demonstrates both rapid calcium return with myocardial relaxation via SERCA2 stimulation and contractility improvement by Na+/K+-AT P-ase inhibition [191]. Therefore, it has also been clinically tested to examine the occurrence of cardiac adverse events, such as arrhythmia [192]. The 24 h infusion has shown beneficial changes in echocardiography parameters, with a lack of major cardiac adverse effects in the acute HF patients’ cohort. These results are in line with the recent study by Metra et al., where the inotropic effect of istaroxime increased cardiac index with blood pressure changes and reduced left ventricular and atrial dimensions [193]. Interestingly, istaroxime shows less cardiotoxic and arrhythmogenic properties alongside significant inotropic effects than classical inotropes [194]. Therefore, it might be reasonably considered a promising agent for HF treatment also with cooperation with mechanical unloading. 

Many studies highlighted the cardioprotective effect of oestrogen receptors (ER) agonists and suggested therapeutic opportunities in HF treatment [195,196]. ER activation mediates several protective pathways, including vasculature, fibrosis, energy metabolism in mitochondria, ROS production, and cardiomyocyte survival. It has been reported that ERβ KO mice with transaortic constriction (TAC)-induced pressure overload have increased cardiac fibrosis and apoptosis in comparison to wild type (WT) or ERβ KO sham surgery mice [197]. Therefore, a lack of ER activation alongside heart injury might, to a greater extent, translate to HF development. Recently, Iorga et al. reported a decreased local heart concentration of oestradiol (E2) and cardiac aromatase transcript levels in the mice model of HF (induced by TAC). Moreover, exogenous treatment of E2 in the same male and female mice improved systolic function, stimulated cardiac angiogenesis, and suppressed fibrosis [198]. Interestingly, the cardioprotective effect of oestrogen has been demonstrated in other animal models of HF [199,200], including right ventricular failure caused by pulmonary hypertension [201]. In light of these results, hormone therapy could represent an interesting direction in HF treatment; however, the lack of knowledge and clinical practice limits its usefulness.

**Table 3 ijms-23-09886-t003:** Overview of ongoing clinical trials focused on either improving LVAD therapy or combining mechanical unloading with pharmacology. Based on clinicaltrials.gov (accessed on 10 June 2022).

Type of Improvement	Study Title	Design	Enrolment/Inclusion Criteria	Agent or Intervention	Time Frame	Primary Outcome	Secondary Outcomes	NCT Number
**Device-related** **modifications**	LVAD Conditioning for Cardiac Recovery	N/A	100/AHF, LVAD implantation as a BTT	Reduction in LVAD speed to minimum operating setting, (during 8 visits, every 2–3 weeks)	Baseline *, 12 month	LVEF	LVEDD, LVESD, LVEDV, LVESV, Glucose 1 and 6-phosphate, Pyruvate, Lactic Acid, Acetyl Coenzyme A, GLUT1, 4, MPC1, 2, mitochondrial density	03238690
	POCT to Improve Monitoring of LVAD Patients	N/A	60/AHF, LVAD implantation	Development of a low-cost detection of LVAD-related coagulation and thrombosis	-	PT/INR LDH	-	03555552
	CYCLONE-LVAD (Role of Cytosorb in LVAD Implantation)	N/A	60/HF, LVAD implantation	Cytokine haemoadsorption by Cytosorb^®^ device to prevent postoperative complication	Baseline *, 6, 12, 24 h, 2, 3, 7 days	IL-6	Prevalence of vasoplegia and organ dysfunction (RV, liver, and kidney), hospitalization, mortality	04596813
	DOAC LVAD (Evaluation of the Hemocompatibility of the Direct OralAnti-Coagulant Apixaban in LVAD)	Phase 2	40/HF, LVAD (HeartMate 3) implantation	Apixaban (5 mg b.i.d.) Warfarin (standard dose and titrated to obtain INR 2.0–2.5.)	3 and 6 month	Survival free of adverse events (stroke, device thrombosis, bleeding, aortic root thrombus), mortality	-	04865978
	The ARIES HeartMate 3 Pump IDE Study	N/A	628/AHF, LVAD (HeartMate 3) implantation	Aspirin (100 mg), vitamin K antagonist (to obtain INR 2.0–3.0)	12, 36 month	Adverse event (stroke, pump thrombosis, bleeding) after 1 year	Rate of survival and adverse events after 3 years	04069156
**LVAD** **with cardiovascular** **drugs**	HARPS ** (Harefield Recovery Protocol Study for Patients With Refractory Chronic Heart Failure)	Phase 1	18/Nonischaemic AHF, LVAD implantation, LVEF ≤ 40%	Clenbuterol (20 tablets mcg t.i.d., titrated to maximally tolerated dose, and then liquid 59 mcg/ml t.i.d)	Baseline *, 2, 6, 12 month	% of LVAD removal and freedom from MCS or HTx for 1 year after explantation	Time to device explant, LVEF, Creatinine, AST, quality of life	00585546
	ENVAD-HF (Sacubitril/Valsartan in LVAD Recipients)	Phase 4	60/LVAD (HeartMate 3) recipients	Sacubitril and Valsartan (24/26 mg, 49/51 mg, 97/103 mg b.i.d.)	Baseline *, 2, 3, and 12 month	Mortality, the occurrence of renal failure, hyperkalaemia, symptomatic hypotension)	BNP, hospitalization, eGFR	04103554
	SEAL-IT (Safety and Efficacy of ARNI After LVAD Implant Study)	Phase 4	50/AHF, NYHA class II-IV and LVEF < 40%	Sacubitril and Valsartan (24/26 mg b.i.d. and increased every 2–4 weeks)	Baseline *, 3, 6, and 12 month	Incidence of medication-related adverse events, BNP	MAP, NYHA class, LVEDD, mitral E/A ratio, LA volume, RAP, PADP, others	04191681
**Mechanical support combined with** **cellular therapy**	LVAD Combined With Allogeneic Mesenchymal Stem Cells Implantation in Patients With End-stage HF	Phase 2, 3	5/AHF due to ischemic cardiomyopathy	Allogeneic stem cells	Baseline *, 12, and 24 month	Myocardial perfusion/viability (SPECT segmental analysis)	Morbidity, LV function	01759212
	ASSURANCE (Stem Cell Therapy in Patients With Severe Heart Failure & Undergoing LVAD Placement)	Phase 1, 2	25/LVAD implantation, NYHA class III or IV, LVEF < 30%, and cardiomyopathy	Bone-Marrow-Derived Mononuclear Cells (20 × 106 cells/400 µL)	Baseline *, 10 weeks, 24 month	Adverse events, Myocardial viability (PET/CT Scan), mortality	LV dimensions, histological assessment	00869024
**Regulation of substrate metabolism** **under** **LVAD condition**	Heart Failure Patients With LVAD Being Treated With Sodium–Glucose Co-Transporter 2 Inhibitors	Phase 4	40/LVAD implantation, eGFR ≥ 30	SGLT2 inhibitors (empagliflozin/dapagliflozin; 10 mg q.d.)	Baseline *, 6 month	LVEDD	-	05278962
**Mitochondria target treatment in LVAD recipient**	PilotNR-LVAD *** (Nicotinamide Riboside in LVAD Recipients)	Early Phase 1	5/AHF, planned elective LVAD implantation	Nicotinamide riboside (1000 mg b.i.d. until LVAD implantation)	-	Incidence of medication-related adverse events	Whole blood NAD+, mitochondrial respiration in isolated peripheral blood mononuclear cells (PBMCs)	03727646
**Immunomodulating** **treatments during** **LVAD support**	Interleukin-1 Receptor Antagonist for the Treatment of Heart Failure in Patients With Left Ventricular Assist Devices	Phase 1, 2	10/LVAD implantation	Anakinra (100 mg SQ q.d. for 2 weeks)	Baseline *, 6 month	CRP	Neutrophil count, EF, TNF-alpha,	02547766

Abbreviations: LVAD: Left ventricular assistance device; N/A: Not applicable; AHF: Advanced heart failure; BTT: Bridge to Transplantation; LVEF: Left Ventricular Ejection Fraction; LVEDD: Left Ventricular End Diastolic Diameter; LVESD: Left Ventricular End Systolic Diameter; LVEDV: Left Ventricular End Diastolic Volume; LVESV: Left Ventricular End Systolic Volume; GLUT: Glucose Transporter; MPC: Mitochondrial Pyruvate Carrier; POCT: Point of Care Testing; PT/INR: Prothrombin Time/International Normalized Ratio; LDH: Lactate Dehydrogenase; IL: Interleukin; RV: Right Ventricular; MCS: Mechanical Circulatory Support; HTx: Heart Transplantation; AST: Aspartate Aminotransferase; BNP: Brain natriuretic peptide; eGFR: estimated glomerular filtration rate; NYHA: New York Heart Association classification; MAP: Mean Arterial Pressure; E/A: early to atrial filling velocity; LA: Left Atrial; RAP: Right Atrial Pressure; PADP: Pulmonary Artery Diastolic Pressure; SGLT2: sodium/glucose cotransporter 2; CRP: C-reactive protein; TNF: tumour necrosis factor. * LVAD implantation, ** Terminated with not fully published results, *** Completed Pilot Study with published results.

## 4. Conclusions

A strategy of combining LVAD with intensive pharmacotherapy has demonstrated more favour in cardiac, clinical, and survival outcomes, with a higher likelihood of myocardial recovery than any other form of HF treatment. In this context, progressively more clinical trials provide credibility to combining the benefits of LVAD therapy with novel pharmacotherapies. Considerable potential exists in stem cell therapy, regulators of substrate oxidation, treatment targeted to mitochondrial biogenesis, structure, and function, as well as inhibitors of a harmful immune response. Despite sustained progress in the engineering, experimental, and clinical fields, further investigation is needed to discover mechanisms of reverse remodelling and increase the effectiveness of the bridge to recovery strategy.

## Figures and Tables

**Figure 1 ijms-23-09886-f001:**
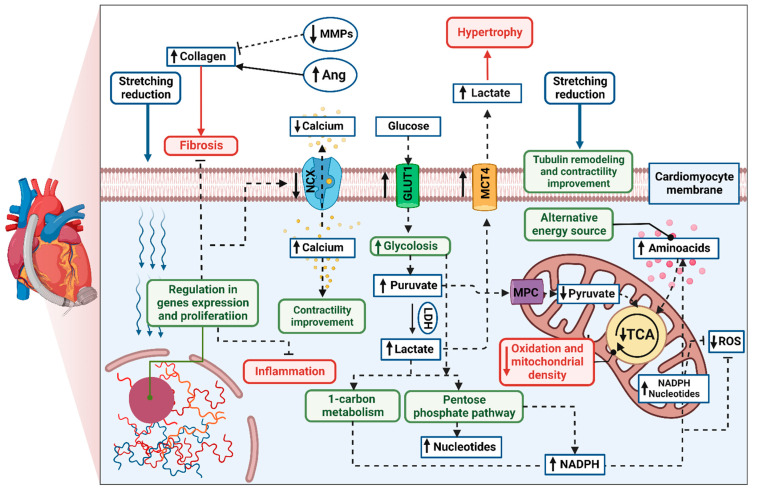
The biochemical status of the mechanically supported cardiomyocyte, where prolonged support exerts up-regulation (↑) or down-regulation (↓) of the metabolite concentrations or receptors/ metabolic pathways activation. The positive (green boxes) and negative changes (red boxes) overlap with each other, limiting cardiomyocyte regeneration but opening the window for novel supportive treatments. Created with BioRender.com (accessed on 2 June 2022).

**Table 1 ijms-23-09886-t001:** Major postoperative adverse events associated with continuous-flow LVAD devices in the medium/long-term.

Adverse Events	Reported Frequency, %	Ref.
Bleeding	30–70	[67]
Right heart failure	20–22	[68]
Haemolysis	18–37	[69]
Acute kidney injury	25–37	[70]
Infection	19–39	[71]
Ventricular arrhythmias	20–50	[72]
Stroke	8–25	[73]
Device malfunction (system failure)	36–51	[74]

**Table 2 ijms-23-09886-t002:** The effectiveness of LVAD therapy in combination with intensive pharmacotherapy.

Study	Treatment	Number	Main Outcomes			
			Cardiac	Clinical	Survival	Recovery *
Yacoub [76]	MU + clenbuterol	17	↑EF, ↓LV volume	↓IL-6, ↓ANP, BNP	N/A	4 (24%)
Birks et al. [78]	MU + LI + CAR+ SPL + LST + clenbuterol	15	↑LVEF, ↓LVED/S Dimension,	↓BNP	91% (1 yr.), 82% (4 yr.)	11 (73%)
Birks et al. [35]	MU + LI + CAR + SPL + LST + DGX + clenbuterol	19	↑EF, ↑FS, ↓LVEDD, ↓LVESD	N/A	83.3% (1–3 yr.)	12 (63.2%)
Grupper et al. [82]	MU	33	↑LVEF (~5%), ↓LVEDD (~4.5%), ↓LVMI (~10%)	↓BNP (~42%)	N/A	N/A
	MU + NHB	31	↑LVEF (~12%), ↓LVEDD (~7.5%), ↓LVMI (~31%)	↓BNP (~88%)	N/A	N/A
Patel et al. [83]	MU + NHB	21	↑LVEF (~90%), ↓LVIDD (~10%), ↓LVMI (~29%)	N/A	N/A	3 (14.3%)
McCullough et al. [84]	MU	1725	N/A	KCCQ = 64.9, 987 ft	44% (4 yr.)	17 (0.99%)
	MU + NHB	10,419	N/A	KCCQ = 68.8, 1103 ft	56% (4 yr.)	169 (1.62%)

* Recovery: LVAD successful explantation due to sustained myocardial recovery. Abbreviations: MU: mechanical unloading; EF: Ejection Fraction; LV: Left Ventricular; IL: Interleukin; ANP: Atrial Natriuretic Peptide; BNP: Brain Natriuretic Peptide; N/A: Not applicable; LI: lisinopril; CAR: carvedilol; SPL: spironolactone; LST: losartan; LVEF: Left Ventricular Ejection Fraction; LVED/S: Left Ventricular End Diastolic/Systolic; DGX: digoxin; FS: Fractional shortening; LVEDD: Left Ventricular End Diastolic Diameter; LVESD: Left Ventricular End Systolic Diameter; NHB: Neurohormonal Blockade; LVMI: Left Ventricular Mass Index; KCCQ: Kansas City Cardiomyopathy Questionnaire; ft: foot; ↑: increased, ↓: decreased.

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
