# Peer review of "Novel Targets for a Combination of Mechanical Unloading with Pharmacotherapy in Advanced Heart Failure"

_ijms, 2022, doi:10.3390/ijms23179886_

Round 1

Reviewer 1 Report

The review is interesting and complete. It covers medical aspects and basic research evidence that allow to understand how cardiac unloading affects both clinical outcomes and biology of the heart. I would recommend to extend just a bit some concepts related to cardiomyocytes function, Ca2+ handling and inotropy. 

One key factor for proper cardiomyocytes function is the development of T tubular sistem, and this has been shown to be regulated by mechanical loading. In fact, unloading can affect T tubules and then systolic calcium release (https://doi.org/10.1093/eurjhf/hfs038). 

The authors mention that SERCA/NCX balance is altered during HF progression and that an improvement in SERCA expression could work togheter with mechanical unloading to preserve cardiac inotropy. It is an interesting point, and given that Ca2+ mishandling can also trigger cardiac arrhythmias it could also reduce the risk of arrhythmia in the focused population. Beyond gene therapy trying to increase SERCA expression, a pharmacological inotrope (Istaroxime) that improves SERCA function could be disscused. It was tested in patients(DOI: 10.1002/ejhf.2629; DOI: 10.1002/ejhf.1743 ) and it was proved to be less cardiotoxic and arrhtyhtmogenic that classical inotropes (DOI: 10.1161/JAHA.120.018833) 

I feel that covering the recommended aspects will improve the manuscript and add some perspective about how novel strategies could improve mechanical unloading strategy. 

Author Response

The review is interesting and complete. It covers medical aspects and basic research evidence that allow to understand how cardiac unloading affects both clinical outcomes and biology of the heart. I would recommend to extend just a bit some concepts related to cardiomyocytes function, Ca2+ handling and inotropy.

We thank the Reviewer for the positive comments in support to our work. We expanded the paper with these points (p.6 l.229-239 and it is also presented on Figure 1).

One key factor for proper cardiomyocytes function is the development of T tubular sistem, and this has been shown to be regulated by mechanical loading. In fact, unloading can affect T tubules and then systolic calcium release (https://doi.org/10.1093/eurjhf/hfs038).

As the Reviewer suggested, this issue has now been explained on p.6 l.229-239, Figure 1. (stretch reduction through mechanical unloading leads to tubulin remodelling and contractility improvement).

The authors mention that SERCA/NCX balance is altered during HF progression and that an improvement in SERCA expression could work togheter with mechanical unloading to preserve cardiac inotropy. It is an interesting point, and given that Ca2+ mishandling can also trigger cardiac arrhythmias it could also reduce the risk of arrhythmia in the focused population. Beyond genetherapy trying to increase SERCA expression, a pharmacological inotrope (Istaroxime) that improves SERCA function could be disscused. It was tested in patients(DOI: 10.1002/ejhf.2629; DOI: 10.1002/ejhf.1743) and it was proved to be less cardiotoxic and arrhtyhtmogenic that classical inotropes (DOI: 10.1161/JAHA.120.018833)

Following the Reviewer's suggestion, we added the additional paragraphs (p.15 l.655-666) that underlined the beneficial effects of SERCA stimulation by Istaroxime in the context to improve the mechanical unloading therapy.

Reviewer 2 Report

This manuscript is a conventional narrative review. It is scientifically sound, clear, comprehensive and presented in a well-structured manner. The paper is relevant for the field, the cited references are current. However, I have identified a small gap. It concerns the subsection on trimetazidine (lines 476-490).

First, the literary citation concerning trimetazidine is incorrect. In the Tuunanen et al. study [130] (line 481), LV EF has increased not by 12.6%, but only by 3.9% (from 30.9 to 33.8%), which is not far from the method error despite the statistical significance of the difference. Furthermore, Fragasso et al. [131] showed an improvement in EF by 7, but not 19%, and a decrease in end-sistolic volume of 17 (correct), not of %, but of ml. Both studies are very small (19 and 55 patients, respectively). Generally speaking, trimetazidine does not have sufficient clinical evidence, which did not allow it to become even a second-line treatment for patients with Heart Failure in the latest ESC Guidelines (2021). It is puzzling to include it in the treatment of chronic coronary syndrome (ECS 2019) as a second line of medicine.

Additionally, the authors' hope for the great perspectives of trimetazidine (“worth exploring” – line 490) is questionable. Trimetazidine has been unable to become an effective cardioprotective agent since it was introduced in 1965. There is no clinical proof of its real beneficial effect in terms of its impact on the cardiac substrate metabolism. As a clinical cardiologist, I would say that trimetazidine has almost no side effects, but its benefits are highly questionable.

Finally, this paper may be recommended for publication after correcting incorrect citations.

Author Response

This manuscript is a conventional narrative review. It is scientifically sound, clear, comprehensive and presented in a well-structured manner. The paper is relevant for the field, the cited references are current. However, I have identified a small gap. It concerns the subsection on trimetazidine (lines 476-490).

First, the literary citation concerning trimetazidine is incorrect. In the Tuunanen et al. study [130] (line 481), LV EF has increased not by 12.6%, but only by 3.9% (from 30.9 to 33.8%), which is not far from the method error despite the statistical significance of the difference. Furthermore, Fragasso et al. [131] showed an improvement in EF by 7, but not 19%, and a decrease in end-sistolic volume of 17 (correct), not of %, but of ml. Both studies are very small (19 and 55 patients, respectively). Generally speaking, trimetazidine does not have sufficient clinical evidence, which did not allow it to become even a second-line treatment for patients with Heart Failure in the latest ESC Guidelines (2021). It is puzzling to include it in the treatment of chronic coronary syndrome (ECS 2019) as a second line of medicine.

Additionally, the authors' hope for the great perspectives of trimetazidine (“worth exploring” – line 490) is questionable. Trimetazidine has been unable to become an effective cardioprotective agent since it was introduced in 1965. There is no clinical proof of its real beneficial effect in terms of its impact on the cardiac substrate metabolism. As a clinical cardiologist, I would say that trimetazidine has almost no side effects, but its benefits are highly questionable.

Thank you for the thorough review of our manuscript and valuable suggestions. We have significantly revised the subsection on trimetazidine (p.12 l.486-502) to address the Reviewer’s comments. We have also changed the incorrect information about the improvement in cardiac function (p.12 l.496-498), added the limitations of clinical trials (p.12 l.498-502), and removed the last sentence.